# Regulation of Ferroptosis in Lung Adenocarcinoma

**DOI:** 10.3390/ijms241914614

**Published:** 2023-09-27

**Authors:** Xiangyun Wei, Xiaohe Li, Shuming Hu, Jinke Cheng, Rong Cai

**Affiliations:** Department of Biochemistry & Molecular Cell Biology, Shanghai Jiao Tong University School of Medicine, Shanghai 200025, China; wxy18759692832@sjtu.edu.cn (X.W.); lxh-2020@sjtu.edu.cn (X.L.); hu-shuming@sjtu.edu.cn (S.H.)

**Keywords:** ferroptosis, regulation, lung adenocarcinoma, therapy

## Abstract

Lung adenocarcinoma (LUAD) is the most common lung cancer, which accounts for about 35–40% of all lung cancer patients. Despite therapeutic advancements in recent years, the overall survival time of LUAD patients still remains poor, especially KRAS mutant LUAD. Therefore, it is necessary to further explore novel targets and drugs to improve the prognos is for LUAD. Ferroptosis, an iron-dependent regulated cell death (RCD) caused by lipid peroxidation, has attracted much attention recently as an alternative target for apoptosis in LUAD therapy. Ferroptosis has been found to be closely related to LUAD at every stage, including initiation, proliferation, and progression. In this review, we will provide a comprehensive overview of ferroptosis mechanisms, its regulation in LUAD, and the application of targeting ferroptosis for LUAD therapy.

## 1. Introduction

Lung cancer is a malignant tumor with the highest morbidity and mortality rate in China and even in the world, of which lung adenocarcinoma (LUAD) is the most common pathological type, accounting for about 40% of lung cancer [1]. LUAD requires individualized treatment based on tumor progression; early-stage LUAD is mainly treated surgically [2], whereas advanced-stage and metastatic patients require systemic therapy [3]. These treatments can significantly improve patients’ symptoms and prolong their overall survival to some extent. However, anti-tumor efficiency in LUAD patients is usually limited by chemotherapy resistance and apoptotic escape, leading to tumor recurrence and a poor prognosis [4,5]. In addition, about 25–30% of lung adenocarcinoma patients obtain KRAS mutations [6]. The small size and smooth surface of the KRAS protein make it hard to target for small molecular drugs [7]. Therefore, the search for specific molecular targets and the development of non-apoptosis-inducing drugs remain a high priority in the treatment of LUAD, especially KRAS mutant LUAD.

Ferroptosis is a novel type of regulated cell death (RCD) with unique molecular alterations and morphological features identified by Dixon et al. in 2012 [8]. The essence of ferroptosis is a membrane rupture and cell death caused by the accumulation of iron-dependent lipid peroxides [9]. There is increasing evidence that application of ferroptosis inducers (FINs) or modulation of ferroptosis-regulated genes (FRGs) can inhibit tumor cell growth and overcome drug resistance due to escape from apoptosis, including LUAD [10]. However, a comprehensive review of the regulation and targeted therapy of ferroptosis in LUAD has not yet been performed. Here, we examine the core mechanisms of ferroptosis and discuss its regulation and therapeutic implications in the context of LUAD.

## 2. Molecular Mechanisms of Ferroptosis

### 2.1. Iron Metabolism

Iron is one of the essential trace elements in the human body and plays important physiological and biochemical functions [11]. Iron in the body generally exists in ferrous (Fe^2+^) or ferric (Fe^3+^) forms, of which Fe^2+^ is essential in inducing ferroptosis. Excess Fe^2+^ will undergo a Fenton reaction with hydrogen peroxide, generating free hydroxide (OH^−^-) and hydroxyl (OH·) radicals that can rapidly promote the peroxidation of membrane lipids, leading to ferroptosis [11]. Iron in normal tissues is in a dynamic equilibrium: circulating Fe^3+^ enters the cell via transferrin (TF) and transferrin receptor (TFR), and intracellular Fe^3+^ is stored as ferritin (including two subtypes, ferritin heavy chain [FHL] and ferritin light chain [FTL]) or reduced to Fe^2+^ and transferred out of the cell to participate in iron recirculation to maintain intracellular iron homeostasis [12,13]. Increasing TF-mediated iron uptake or NCOA4-mediated ferritinophagy leads to an increase in intracellular labile iron. Ferritinophagy refers to the specific recognition of and binding to ferritin by nuclear receptor coactivator 4 (NCOA4), which mediates the autophagic degradation of ferritin in the lysosome, thereby promoting the release of Fe^2+^ [14]. The expression of NCOA4 can promote erastin-induced ferroptosis in LUAD cells by promoting FHL autophagy [15].

### 2.2. Lipid Metabolism

Another important factor for inducing ferroptosis is the peroxidation of polyunsaturated fatty acid-containing phospholipids (PUFA-PLs), especially arachidonic acid (AA) and adrenic acid (AdA) [16]. The formation of lipid peroxides needs three key enzymes: acyl-CoA synthetase long-chain family member (ACSL4), lys phosphatidylcholine acyltransferase 3 (LPCAT3), and lipoxygenases (ALOXs) [16,17]. Among them, ACSL4 is responsible for catalyzing the acylation between AA/AdA and CoA to generate AA-CoA/AdA-CoA, while LPCAT3 is responsible for catalyzing the re-esterification of AA/AdA-CoA and membrane phosphatidylethanolamines (PLs) to generate AA/AdA-PE, and the ALOXs, a class of enzymes with iron as a cofactor, are responsible for oxidizing PUFAs to produce hydroperoxyl derivatives such as 4-malondialdehyde (MDA) and hydrodynamical (HNE). Common ALOXs are ALOXE3, ALOX5, ALOX12, ALOX12B, ALOX15, and ALOX15B [17]. Moreover, Cytochrome P450 oxidoreductase (POR) can also mediate lipid peroxidation and promote ferroptosis [18]. Notably, ACSL4 and LPCAT3 were expressed at elevated levels in highly malignant LUAD patients [19], whereas 4-HNE was lower in highly staged tumors relative to low-stage tumors and normal tissues [20], all of which are associated with high ferroptosis sensitivity, suggesting that lipid metabolism modulates the ferroptosis sensitivity in LUAD in a multifaceted and multi-mechanistic manner. On the other hand, monounsaturated fatty acids (MUFAs), synthesized by stearoyl-CoA desaturase1 (SCD1), can competitively bind to cell membranes, thereby inhibiting ferroptosis [19].

### 2.3. Antioxidant Defenses

Cell defenses against ferroptosis are mediated by multiple antioxidant systems, such as the solute carrier family 7 member 11 (SLC7A11)-reduced glutathione (GSH)-glutathione peroxidase 4 (GPX4) pathway [21], apoptosis-inducing factor mitochondria-associated 2 (AIFM2, recently renamed as ferroptosis suppressor protein 1 (FSP1))-coenzyme Q (CoQ10) axis [22], and the GTP cyclohydrolase-1(GCH1)-tetrahydrobiopterin/dihydrobiopterin (BH4/BH2) pathway [23].

Among them, the SLC7A11-GSH-GPX4 pathway is the most important and earliest discovered antioxidant system [21]. SLC7A11 is a cystine/glutamate transporter that can uptake extracellular cystine into the cell and transport glutamate out of the cell simultaneously [24]. And the cystine that enters the cell can be used for the synthesis of GSH, which is not only an important antioxidant but also a preferred substrate for GPX4. GPX4, a selenoprotease, can act as a detoxifier by converting highly toxic lipid hydroperoxides into non-toxic lipiodols and hydroperoxides into water. Erastin, a ferroptosis inducer, directly inhibits the activity of the X_C_ system, leading to the accumulation of intracellular glutamate, a decrease in GSH synthesis, and a reduction in the activity of GPX4, thereby leading to cell ferroptosis [8]. Thus, SLC7A11 and GPX4 are important ferroptosis regulators as well as potential targets for cancer therapy.

Next, the FSP1-CoQ10 pathway was recently found to be able to protect cells from ferroptosis by reducing CoQ10 to produce ubiquinol, a lipid radical trapping agent [22]. Notably, FSP1-CoQ10 functions in a subcellular organelle-specific manner, with pro-apoptotic effects in mitochondria and anti-ferroptosis in the plasma membrane. In non-small cell lung cancer (NSCLC), colorectal cancer (CRC), and pancreatic ductal adenocarcinoma (PDAC), FSP1 is expressed at higher levels in KRAS mutant tumor cells compared to normal cells [25]. But considering the diversity of FSP1 functions, it is reasonable to believe that FSP1 may also promote tumor progression through mechanisms other than ferroptosis. Finally, the GCH1-BH4/BH2 pathway can function as an antioxidant, protecting against ferroptosis in the absence of GPX4 [23].

## 3. Ferroptosis in LUAD

### 3.1. Ferroptosis in LUAD Tumorigenesis and Progression

#### 3.1.1. Ferroptosis in LUAD Tumorigenesis

About 25–30% of lung adenocarcinomas develop KRAS mutations, with the KRAS-G12C point mutation being the most common [6,26]. When the G12C mutation occurs in *KRAS*, its GTP hydrolase activity will be lost, which will activate several signaling pathways such as RAF-MEK-ERK, PI3K-AKT-mTOR, and Ral-GDS. The cells will then have malignant potential and gradually become cancerous, which will eventually lead to the development of LUAD [27]. It is established that KRAS-mediated cellular transformation requires the generation of reactive oxygen species (ROS) due to elevated expression of NADPH oxidase 1 (Nox1) [28]. So how do cells expressing mutant KRAS alleviate ROS-induced cell death? That is, FSP1 expression was sufficient in KRAS-mutated LUAD to significantly promote 3D spheroid growth in vitro and accelerate tumor onset in vivo by protecting from ferroptosis [25]. Consistent with this finding, Zhang et al. showed that KRAS-driven LUAD has a greater resistance to ferroptosis owing to a reprogrammed lipid metabolism by a higher level of acyl-coenzyme A synthetase long-chain family member 3 (ACSL3) expression [29]. Due to the small size and smooth surface of the KRAS protein, there is no suitable binding pocket for other small molecules except the one for GTP/GDP, and the affinity of RAS for GTP is very high, which makes it difficult for drugs to compete with the substrate [7]. The treatment of KRAS-mutated LUAD has been a challenge. Intervention of FSP1 and expression in the early stage of atypical adenomatoid hyperplasia (AAH) to adenocarcinoma in situ (AIS) or to micro-infiltrating adenocarcinoma (MIA), and thus intervention in ferroptosis of tumor initiation, may be a novel therapeutic direction for KRAS-mutated LUAD (Figure 1).

#### 3.1.2. Ferroptosis in LUAD Progression

Ferroptosis is also linked to the growth and development of LUAD. Researchers found many ferroptosis inhibitors act as indicators of poor prognosis and promote tumor cell growth in LUAD, such as CCT3 and NFS1 [30,31]. In addition, bioinformatics analyses have revealed a potential clinical connection between the FRGs and LUAD patients. A prognostic model integrating several FRGs was used to predict the prognosis, mutation burden, tumor microenvironment (TME) cell infiltration characteristics, immunotherapy effects, and chemotherapy sensitivities in patients with LUAD [32,33,34,35,36,37].

Epithelial-mesenchymal transition (EMT) and ferroptosis are two important processes in tumor progression, and recent studies reported that they may form a positive feedback loop to a certain extent in LUAD. On the one hand, it has been known that highly mesenchymal-like tumor cells are indeed more sensitive to ferroptosis inducers. E3 Ligase MIB1 can promote EMT while at the same time promoting ferroptosis through NRF2 degradation [38]. ZEB1 is one of the major transcription factors that regulates EMT by binding to the E-box in E-cadherin [39]. In LUAD, ZEB1 correlates with the transcription of LPCAT3 and thus increases the susceptibility to ferroptosis [19]. On the other hand, the ferroptosis tendency also enhances the development of EMT. Erastin, which is an inducer of ferroptosis, can reduce the expression of E-cadherin and cause de-epithelialization. Ferrostatin-1 (Fer-1), an inhibitor of ferroptosis, can partially inhibit EMT induced by TGF-β1 [40]. Not coincidentally, some ferroptosis markers (GPX4, SCP2, and CAV1) again suggest that ferroptosis can positively regulate the occurrence of EMT in vivo [19]. However, there is also a negative feedback loop between EMT and ferroptosis. High ARNTL2 expression was associated with EMT and lymph node metastasis in patients with LUAD, while it plays an inhibiting role in ferroptosis [41].

Epidermal growth factor receptor (EGFR) is the most common mutation in LUAD, with a prevalence of 15% of Caucasians and 50% of Asians [42,43]. EGFR tyrosine kinase inhibitors (EGFR-TKIs) are used to treat EGFR mutant LUAD. First- to third-generation EGFR-TKIs have been approved both domestically and internationally, and fourth-generation EGFR-TKIs such as BLU-945 have entered clinical studies [44,45]. EGFR-mutant LUAD cells also increase cellular sensitivity to ferroptosis. Low-dose selenite synergized with osimertinib in EGFR-mutant H1975 [46]. Acquired resistance is still inevitable with the use of EGFR-TKIs, and about 20–30% of EGFR mutant LUADs are intrinsically resistant to EGFR-TKIs [47,48,49]. In addition to the mesenchymal state and EGFR-mutant tumor cells mentioned above, EGFR-TKI-resistant LUAD cells also increase cellular sensitivity to ferroptosis, and the histone deacetylase inhibitor Vorinostat can further downregulate the expression of xCT in EGFR mutant LUAD cells and enhance the effect of ferroptosis induction therapy [50]. Further work on the clinical effect of these drugs and their combination with TKIs on EGFR mutation-LUAD is warranted. Moreover, chemoresistance also makes LUAD cells more sensitive to ferroptosis, and promoting ferroptosis can overcome or reverse the resistance of tumor cells to cisplatin, pemetrexed, and Lapatinib [51,52,53].

TME constitutes the balance between tumor cells and immune cells and can have both adverse and beneficial consequences during tumor progression. Ribonucleotide reductase subunit M2 (RRM2) not only affects tumor cell proliferation but also regulates immune cell infiltration, thereby influencing lung cancer progression in a ferroptosis-dependent manner. For one thing, depletion or silencing of RRM2 inhibited proliferation and induced ferroptosis in H1975 and H358 LUAD cells [54,55]. For another, RRM2 effectively promoted M2 macrophage polarization, facilitating tissue repair and LUAD development in vitro and in vivo [56]. RRM2 also regulated the infiltration levels of activated mast cells and activated CD4 memory T cells, again suggesting that RRM2 may be engaged in immune infiltration [54]. Not coincidentally, bioinformatics has demonstrated that other FRGs can modify the behavior of TME cells and that these subtypes of TME cells exhibit distinct biological features and communicate extensively with tumor epithelial cells (Figure 1). Patients with a higher abundance of these ferroptosis-related TME cell subtypes have a better clinical outcome [53]. In addition, ferroptosis-related prognostic signatures, such as the prognostic ferroptosis-related lncRNA signature and GPX4-related prognostic signature, are not only correlated with multiple tumor-infiltrating immune cells and immune-associated processes and pathways in TME, but also with the response to immunotherapy, chemotherapy, and targeted therapy [57,58].

### 3.2. Regulation of Ferroptosis in LUAD

Ferroptosis is essentially a form of cell death caused by oxidative damage. Like other RCDs (e.g., apoptosis, pyroptosis, and necrosis), it is modifiable. In LUAD, the core regulators of ferroptosis are mainly regulated at the transcriptional and epigenetic levels (Figure 2).

#### 3.2.1. Transcriptional Regulation

In LUAD, transcriptional factors and cofactors such as NRF2, TP53, YAP, ARNTL2, CREB, KLF11, etc. are key regulators in various steps of ferroptosis and thus in tumor pathogenesis and progression.

##### NRF2

The master transcriptional factor nuclear factor erythroid 2-related factor 2 (NFE2L2, also known as NRF2) plays an important antioxidant role in maintaining redox homeostasis. The role of NRF2 in ferroptosis is a popular subject of ongoing research, with a majority of studies suggesting that NRF2 plays an anti-ferroptotic role in LUAD cells. As we know, under the induction of oxidative stress, NRF2 can transfer into the nucleus, where it binds to partner proteins such as Maf and subsequently initiates the transcription of a series of antioxidant genes, including NAD(P)H quinone dehydrogenase 1 (NQO1) [52], GPX4, SLC7A11, the aldo-keto-reductase-1C (AKR1C) family [59], and heme oxygenase-1 (HO-1), as well as FSP1 [25]. In KRAS mutant cells, the *Aifm2* mRNA, encoding the protein FSP1, is upregulated because of MAPK-NRF2 pathway activation [25]. The expression of FSP1 can protect cells from ferroptosis-associated lipid ROS accumulation [22,60]. Although this is contrary to the previous finding that overexpression of oncogenic KRAS in fibroblasts resulted in sensitization to erastin-induced ferroptosis [61]. It coincides with the fact that GPX4 deletion in pancreatic intraepithelial neoplasia (PanINs) is not sufficient to trigger ferroptosis in genetically engineered mouse models, supporting the idea that KRAS mutant cells have evolved additional antioxidant mechanisms to counteract ferroptosis [62]. Among them, NRF2-mediated transcriptional activation may be critical in determining whether oncogenic KRAS expression is pro-ferroptosis or anti-ferroptosis. Up to half of patients present with activating KRAS mutations [25]. It is therefore tempting to speculate whether combined induction of ferroptosis and inhibition of FSP1 can increase the therapeutic effectiveness of KRAS-mutated LUAD in a clinical setting.

Ubiquitination and proteasome-mediated degradation are the most important mechanisms regulating the expression and activity of NRF2, which in turn controls the ferroptotic resistance to LUAD. First and most important is the Keap1-cullin 3 (Cul3)-E3-mediated ubiquitinated protein degradation [63]. Kelch-like ECH-associated protein 1 (KEAP1), which maintains low levels of NRF2 in homeostasis, mediates 26S proteasome degradation by recruiting a Cul3-containing E3 ubiquitin ligase complex to NRF2 [63,64]. KEAP1-mutant lung adenocarcinoma accounts for about 17% of LUAD [65]. Inactivation of KRAP1 leads to constituent activation of NRF2 signaling, which favors tumorigenesis, so that even with the intervention of chemotherapy and immunotherapy, the survival of such LUAD patients is shorter [66,67]. For example, activation of the KRAP1-NRF2-AKRs pathway can maintain the proliferation rate of LUAD cells, where AKR1C1/2/3 antagonizes ferroptosis by promoting the detoxification of active intermediates of aldehydes and ketones. Among others, this effect is greatly enhanced by the cooperativeness of STK11 and KEAP1 loss-of-function, independent of KRAS mutation status [59,66]. It was found that, in addition to KEAP1, NRF2 is also capable of being degraded by MIB1, another E3 ubiquitin ligase [38]. High expression of MIB1 in A549 cells affects HMOX1 transcription and subsequent lipid ROS clearance by downregulating NRF2. Lung squamous and adenocarcinoma patients with high MIB1 expression are also associated with lower survival rates. Vice versa, ubiquitination of NRF2 can be removed by de-ubiquitinating enzymes (DUBs) because ubiquitination is a dynamic and reversible process, and this process in non-small cell lung cancer (NSCLC) is mediated by ubiquitin-specific processing protease 11 (USP11) [63]. Our study revealed that RSL3 promotes ferroptosis in A549 and H2122 cells by directly inhibiting USP11-mediated de-ubiquitination and promoting degradation of NRF2, thereby repressing transcription of SLC7A11 and GCL [68].

##### p53

Wild-type TP53 is a suppressor that negatively regulates the growth of LUAD, while mutant TP53 has oncogenic activity and is one of the critical causes of LUAD transformation. Approximately 33% of LUAD patients are associated with mutations in the *TP53* gene, and Thompson et al. reported that LUAD NCI-H1299 cells expressing mutant p53 had increased sensitivity to the ferroptosis inducer, erastin, compared to WT p53 cells [69]. This may be due to its suppression of SLC7A11 expression through entrapping the transcription factor NRF2 [70]. In addition, p53 R273H cells were more vulnerable to AF-induced ferroptotic cell death due to the downregulation of GPX4 [71]. Such a reduction in SLC7A11 and inactivation of GPX4 facilitates the inhibition of LPO elimination and elevation of ROS levels, which amplifies oxidative stress and promotes ferroptosis. This is also the working principle of metal-organic nanodrug p53/Ce6@ZF-T, which was composed of p53 plasmid-complexed chlorin e6 (Ce6)-poly(amidoamine), Fe^2+^-containing mesoporous zeolitic imidazolate framework-8 and naturally derived tannic acid (TA). Highly cytotoxic ROS were continuously produced via Fe^2+^-mediated OH. generation and Ce6-catalytic ^1^O2 production, while restoration of p53 would significantly enhance -OH and ^1^O2 accumulation, leading to amplified oxidative stress and enhanced ferroptosis therapy [72]. Acetylation is an important layer of the p53 functional regulation mechanism in ferroptosis and tumor inhibition. In fact, mouse p53 3KR mutations at K117, K161, and K162 activated ferroptosis and inhibited tumor growth, but the additional mutation of p53 at K98 would abolish this ferroptosis-promoting process [73]. Additionally, GINS4 suppressed p53-mediated ferroptosis through activating Snail, which antagonized the acetylation of p53 lysine residue 351 (K351 for human p53), thereby resulting in an upregulation of the ubiquitination level and a decrease in the stability of p53 [74].

##### YAP

YAP, a cotranscription factor acting downstream of the Hippo pathway, not only functions as a proto-oncoprotein, but is also closely related to ferroptosis sensitivity in LUAD [75]. Inhibition of system Xc activity, which reduces cystine entry into the cell, leads to intracellular glutamate accumulation simultaneously. As previously mentioned, ferroptosis sensitivity varies widely in cells and tissues and may be regulated by the release of liable iron and lipid peroxides [14]. Zhang et al. first found that erastin inhibition of system Xc-induced endogenous glutamate accumulation was able to determine ferroptosis sensitivity of LUAD cells via inhibiting the ADCY/PKA/HBP/YAP axis [15]. Uridine-diphosphate N-acetylglucosamine (UDP-GlcNAc), a donor for O-GlcNAcylation, is synthesized from glucose through glutamine-fructose-6-phosphate transaminase (GFPT1) [76]. Glutamate accumulation promotes ADCY10-catalyzed cAMP generation in a Ca^2+^-dependent manner, and then GFPT1 is phosphorylated and inhibited by cAMP-dependent PKA [15]. Thereby, the O-GlcNAcylation and stability of YAP are reduced. YAP is a coactivator that does not function as a direct transcription factor and requires the formation of a transcription-promoting complex with other transcription factors [77]. Next, they also found that YAP regulates iron-dependent ferroptosis sensitivity through the formation of the YAP-TFCP2-complex [15,20]. Inhibition of YAP promotes NCOA4-mediated ferritinophagy in different LUAD cell lines [15]. And inhibits FTL transcription via the YAP-TFCP2-FOXA1 complex in PC9 and H1299 cell lines, thus doubly promoting labile iron production to enhance the ferroptotic effect [20]. Finally, Zhang et al. also found that YAP regulates lipid-dependent ferroptosis sensitivity through the formation of the YAP-ZEB complex. As previously stated, the lipid metabolism-related enzyme LPCAT3 is a ferroptosis inducer [78]. The YAP-ZEB complex binds to the −1600 to −1401 nt LPCAT3 promoter, promoting its transcription, lipid peroxidation, and ferroptosis [19]. These studies have not only identified therapeutic targets of ferroptosis (such as YAP, FTL, and FTH), but have also identified several ferroptotic sensitivity biomarkers (such as CODY10 and LPCAT3) in LUAD, all of which may provide support for ferroptosis use in clinical oncology therapy.

##### Others

Ferroptosis is, in fact, an iron-dependent, unchecked lipid peroxidation form of cell death [8]. In addition to the regulation on the ferroptosis by NRF2 and YAP mentioned above, any other transcription factors capable of influencing the transcription of lipid oxidation and anti-lipid oxidation genes would be able to intervene in ferroptosis. As a first example, the aryl hydrocarbon receptor nuclear translocator like 2 (ARNTL2), as a circadian transcription factor, increases the expression of acyl-CoA thioesterase 7 (ACOT7) via binding to the *ACOT7* promoter, which in turn promotes the synthesis of MUFAs such as oleic acid and palmitoleic acid [79]. Given that ARNTL2 promotes EMT and cell proliferation and migration in addition to inhibiting lipid peroxidation and ferroptosis via ACOT7, ARNTL2 may be a poor biomarker for LUAD [41]. As a second example, cAMP response element-binding protein (CREB) and kruppel-like factor 11 (KLF11) can promote or repress GPX4 expression by binding to the *GPX4* promoter, respectively. The high expression of CREB in LUAD patients’ tumor tissues and KLF11 in RSL3/IKE-treated A549 and PC9 cells further supports the idea that CREB or KLF11 are positive or negative regulators of antioxidant defense [80,81].

#### 3.2.2. Epigenetic Regulation

In addition to ferroptosis regulation at the transcriptional level, the susceptibility of LUAD to ferroptosis can also be regulated at several epigenetic levels, including DNA methylation, histone modification, and various non-coding RNA-mediated processes. Despite recent emerging evidence supporting a role for DNA methylation in regulating GPX4 expression and the widespread use of ferroptosis-related DNA methylation signatures as predictive and prognostic biomarkers for a wide range of cancers [82,83,84], DNA methylation in the ferroptosis sensitivity of LUAD remains relatively understudied. For example, a recent study mentions that selenite is a potent LUAD FIN along with activating DNA methylation. However, selenite induction of ROS and possible inhibition of xCT expression, as well as downregulation of DNMT1-mediated TET1 DNA methylation, are two parallel antineoplastic mechanisms in EGFR- and potentially KRAS mutant lung cancer cells (H1975 and H358 cell lines), and there is no mention of an interconnection between the two [46]. So, we next focus on the roles of two other epigenetic mechanisms in the regulation of LUAD ferroptosis (Table 1).

##### Histone Modification

As we all know, histone modifications such as methylation and acetylation can regulate transcriptional activity [85]. The acetyltransferase E1A binding protein P300 (EP300), for example, promotes the binding of transcription factors to target genes by increasing H3K27ac [86]. Most typically, EP300 binds to the bZIP domain of CREB via its CBP/p300-HAT domain to form a transcription-promoting complex [87]. This transcriptional complex greatly enhances the binding of CREB to the *Gpx4* promoter in the previously mentioned A549 and H1299 cells [80]. EP300 can form a ternary complex with YAP and ZEB, wherein the YAP WW domain interacts with the ZEB ZF domain, and YAP also binds to the Bromo domain of EP300, while ZEB binds to the CBP/p300-HAT domain of EP300 [19]. Given that the ternary complex formed by two-by-two binding of the three proteins is much more stable than the binary complex, targeting this EP300-YAP-ZEB transcription complex may be a new strategy for treating LUAD.

##### Non-Coding RNA-Mediated Processes

miRNAs

An increasing number of miRNAs, small (17–24 nucleotides) non-coding RNAs, have been recognized as important players in ferroptosis regulation. They can bind to the 3’-UTR of messenger RNAs (mRNAs) and mediate the post-transcriptional silencing of FRGs [88]. For example, miR-27a-3p regulates SLC7A11 expression to alter the sensitivity of A549 and Calu-3 LUAD cells to erastin-trigged ferroptosis [89]. miR-324-3p regulates GPX4 to alter the desensitization of A549 DDP cells to cisplatin-induced ferroptosis [51], and bioinformatics analyses predicted that SLC7A11 and GPX4 are targets of miRNAs such as has-mir-37 and miRNA-126-3p/5p [90]. But whether and how these miRNAs regulate them deserves further investigation.

CircRNAs

Regarding miRNAs, it is important to mention competing endogenous RNAs (ceRNAs), generally circRNAs and lncRNAs, which act as “miRNA sponges” to mitigate the inhibitory effects of miRNAs on their target genes [91,92]. Pan et al. added another example of this in LUAD. They found that circRNA P4HB acts as a sponge for miRNA-1184 and competitively binds miRNA-1184 to SLC7A11, thereby inhibiting its inhibitory effect on *Slc7a11* and promoting *Slc7a11* expression and subsequent GSH synthesis [93]. This circP4HB/miR-1184/SLC7A11 axis protects LUAD from erastin-induced ferroptosis and promotes tumor growth in vivo and in vitro. In addition to functioning as ceRNAs, circRNAs also directly interact with proteins to inhibit ferroptosis in LUAD [94]. The exosome-derived circRNA, circ_1010093, acts as a novel ferroptosis suppressor in LUAD. Autocrine cir93 secreted by tumor cells inhibits AA incorporation into the plasma membrane and reduces lipid peroxidation in a FABP3-dependent manner. Mechanistically, the exosomal cir93 maintains intracellular levels of cir93 and promotes its binding to the AA transporter protein fatty acid-binding protein 3 (FABP3), thereby reducing AA global levels via the reaction of AA with taurine. Moreover, N-arachidonoyl taurine (NAT), the product of AA and taurine, inhibits ACSL4, LPCAT3, and PLTP expression by competitive binding to transcription factor TAX2, thereby further reducing AA incorporation into the plasma membrane [95]. This shows that cellular communication (autocrine secretion of cir93) in the tumor microenvironment is also capable of activating the antioxidant system and inhibiting ferroptosis. Both circP4HB in tumor cells and cir93 in patient plasma are biomarkers of LUAD and deserve further investigation in the early diagnosis and treatment of LUAD.

LncRNAs

Another typical ceRNA is long non-coding RNAs (lncRNAs), a large number of which act as oncogenes and against cellular ferroptosis in LUAD, including the LINC00324/miR-200c-3p/TFAP2A pathway [32], the lncRNA GSEC/miR-101-3p/CISD1 pathway [96], the LncRNA T-UCR Uc.339/miR-339/SLC7A11 pathway [97], and the LINC00336/miR-6852/CBS pathway [98]. Of these, LINC00336 is more specific. Not only is LINC00336 able to regulate miR-6852, but miR-6852 is also able to inhibit LINC00336 in turn. Furthermore, this ferroptosis resistance is not solely dependent on a simple lncRNA-miRNA interaction. There is also a lncRNA-protein interaction between LINC00336 and ELAV-like RNA-binding protein 1 (ELAVL1). Binding of ELAVL1 to LINC00336 prolongs its half-life, thereby further enhancing its resistance to ferroptosis [98]. There are also some lowly expressed lncRNAs in LUAD, such as the ferroptosis promotor and tumor suppressor. As the first example, GMDS-AS1 and LINC01128 specifically reduce miR-6077 and sensitize LUAD cells to cell cycle arrest and ferroptosis [52,99]. A possible explanation is that miR-6077 interacts with and suppresses CDKN1A and KEAP1, thereby inhibiting CDKN1A-mediated G2/M arrest and KEAP1-NRF2-SLC7A11/NQO1-mediated ferroptosis. Thus, GMDS-AS1 and LINC01128 can restore the sensitivity of LUAD to cisplatin/pemetrexed combination chemotherapy [52]. As the second example, LINC00551 promotes ferroptosis of LUAD cells in an autophagy-dependent manner by upregulating DDIT4 via competitively binding miR-4328. DNA damage-inducible transcript 4 (DDIT4) can inhibit the mTOC1 complex, thereby promoting autophagy and ferroptosis [99]. This is consistent with previous findings that DDIT4 is a driver of autophagy-dependent ferroptosis in pancreatic cancer cells [100].

Collectively, the above non-coding RNAs associated with ferroptosis may be novel targets or biomarkers for LUAD prevention and treatment.

m6Amodification

N6-methyladenosine (m6A) is a WER system consisting of writers (W), erasers (E), and readers (R). Writers are methyltransferases, while erasers are enzymes that catalyze demethylation [101]. Unlike writers and erasers, readers function as RNA-binding proteins that recognize m6A modifications and regulate mRNA splicing, export, translation, and stability [102]. Emerging research shows that m6A modification and its associated proteins are significant regulators of ferroptosis and LUAD tumorigenesis [103,104]. For example, methyltransferase-like 3 (METTL3) desensitizes ferroptosis by enhancing the stability and translation of SLC7A11 mRNA via recruiting TYHDF1 to SLC7A11 m6A modification [105]. The m6A reader insulin-like growth factor 2 mRNA binding protein 3 (IGF2BP3) is identified as an ferroptosis inhibitor via sustaining m6A-methylated mRNAs encoding anti-ferroptosis factors, including GPX4, SLC3A2, ACSL3, and FTH1 [106]. Both METTL3 and IGF2BP3 are highly expressed in LUAD patients and associated with a poor prognosis [105,106]. Moreover, the m6A reader YT521-B homology containing 2 (YTHDC2) was identified as an endogenous ferroptosis inducer and can inhibit LUAD tumorigenesis [107]. Mechanically, the two subunits of system Xc, both SLC7A11 and SLC3A2, can be suppressed in a m6A-dependent manner. YTHDC2 was able to accelerate the degradation of SLC7A11 mRNA by directing binding to the 3’UTR of SLC7A11 mRNA. YTHDC2 was also able to indirectly inhibit SLC3A5 expression by destabilizing the SLC3A5 transcription factor HOXA13 mRNA. Therefore, increasing YTHDC2 in LUAD patients may be an attractive and alternative ferroptosis-targeted therapy [107].

**Table 1 ijms-24-14614-t001:** Epigenetic regulations of ferroptosis in LUAD.

Name	Epigenetic Mechanism	Functions in Ferroptosis	Expression in LUAD Patients	References
EP300	histone modification	① EP300/CREB complex: suppressor; increases GPX4 expression ② EP300/YAP/ZEB complex: promoter; increases LPCAT3 expression	N/A	[19,79]
miR-27a-3p	miRNA	Promoter; inhibits SLC7A11 expression	Downregulation	[88]
miR-324-3p	miRNA	Promoter; inhibits GPX4 expression	Downregulation in A549/DDP cells	[51]
P4HB	circRNA	Suppressor; sponge miRNA-1184 and increases SLC7A11 expression	Upregulation	[92]
circ_1010093	circRNA	Suppressor; inhibits lipids (ACSL4, LPCAT3, and PLTP)	Upregulation	[94]
LINC00324	lncRNA	Suppressor; sponge miR-200c-3p and promotes TFAP2A-NRF2 axis	Upregulation	[37]
GSEC	lncRNA	Suppressor; sponge miR-101-3p and increases CISD1 (a mitochondrial iron–sulfur protein)	Upregulation	[95]
Uc.339	lncRNA	Suppressor; sponge pri-miR-339 inhibits the production of mature miR-339 and increases SLC7A11 expression	Upregulation	[96]
LINC00336	lncRNA	Suppressor; sponge miR-6852 and increases cystathionine-β-synthase (CBS, involved in the transsulfuration pathway and synthesizes cysteine) expression	Upregulation	[97]
GMDS-AS1 and LINC01128	lncRNA	Suppressor; sponge miR-6077 and promotes the KEAP1-NRF2-SLC7A11/NQO1 pathway	Downregulation	[59]
LINC00551	lncRNA	Promotor; sponge miR-4328 and increases DDIT4 expression; DDIT4 inhibits mTOR activity and promotes autophagy-dependent ferroptosis	Downregulation	[98]
METTL3	m6A	Suppressor; enhance the stability and translation of SLC7A11	Upregulation	[104]
IGF2BP3	m6A	Suppressor; enhances the stability and translation of anti-ferroptotic factors (GPX4, SLC3A2, ACSL3, and FTH1)	Upregulation	[105]
YTHDC2	m6A	Promotor; inhibits system Xc (directly inhibits SLC7A11 and indirectly inhibits SLC3A5 expression)	Downregulation	[106]

### 3.3. Ferroptosis in LUAD Therapy

Targeting ferroptosis has the potential to be a prospective strategy for combating LUAD, especially in chemotherapy-resistant LUAD patients. Combinatory therapy with ferroptosis inducers and chemotherapeutic agents may produce improved therapeutic effects in LUAD patients with Cisplatin/sorafenib/Osimertinib resistance [51,52]. In addition, small molecular compounds like sorafenib [108,109], Chinese medicines [110,111], natural products [112], and nanoparticles [113] are showing as potential ferroptosis inducers in LUAD.

The multi-targeting properties of Chinese medicine have attracted research into whether it can modulate ferroptosis in LUAD. Tetrandrine (Tet) is isolated from the plant *Stephania tetrandra*, *S. Moore,* and has been widely used in pulmonary fibrosis and anti-tumor therapy [114]. It has also been identified as a potential FIN that can inhibit GPX4 by suppressing the expression of SLC7A11 [110]. Mixing Tet with citric acid in ddH_2_O at a ratio of 4:1 produces Tetrandrine Citrate (TetC), which also overcomes the hydrophobicity of Tet that makes it difficult to form drugs. Moreover, Hedyotis diffusa, another antitumor effector, is also able to induce ferroptosis by inhibiting Bcl2 expression, promoting Bax expression, and thus promoting the activation of VDAC2/3 [111]. Voltage-dependent anion channel 2/3 (VDAC2/3) is a mitochondrial channel protein whose sustained activation promotes mitochondrial depolarization, facilitates ROS release [115,116], and induces ferroptosis. Previous studies have shown that erastin can also induce ferroptosis through activation of VDAC2/3 [117]. Both TetC and VDAC2/3 have the advantage of low toxicity, with little drug damage to the heart, liver, and kidneys. Combining natural products like Ginkgetin with cisplatin can enhance the anticancer effect in NSCLC. Ginkgetin derived from Ginkgo biloba leaves induces ferroptosis by disrupting the Nrf2/HO-1 antioxidant system, increasing ROS release and mitochondrial membrane potential loss [112]. Moreover, an inhalable nanoreactor was proposed to enhance LUAD ferroptosis therapy [113].

## 4. Conclusions and Perspective

The discovery of ferroptosis has provided imaginative insights into the treatment of LUAD and overcoming drug resistance. In recent years, the study of ferroptosis in LUAD has grown considerably, as have many other signaling pathways, such as FSP1-CoQ10 and dihydroorotate dehydrogenase (DHODH)-CoQ10, which have been found to regulate ferroptosis in addition to the classical SLC7A11-GPX4 antioxidant pathway. These signals are potential anti-tumor targets that could be used to complement FIN therapies based on small-molecule compounds such as erastin and RSL3. However, there are also many challenges to the pharmacology of these inhibitors. Firstly, most of the inhibitors act via one-point to one-point. As we know, excessive lipid peroxidation, unbalanced redox homeostasis, and labile iron overload are three major hallmarks of ferroptosis. The ideal, cost-effective strategy to induce ferroptosis is to simultaneously inhibit these factors. Whether a single target is sufficient to induce ferroptosis and exert clinical efficacy is worth investigating, as the present study demonstrated that the oncogenic effect of MPA, a pan-inhibitor targeting AKR1Cs (a family of ferroptosis-protective genes), was not significant enough to be used as a therapeutic strategy [66]. In addition, the pharmacological properties of many FINs are not well understood, and the possible side effects of their clinical application have yet to be investigated. Finally, the specificity of FINs is also an issue, and FINs that attack tumor cells and normal cells indiscriminately have no clinical value.

Research on the regulation and application of ferroptosis in LUAD is still in its infancy. There are still many aspects worth exploring or deepening, such as the link between ferroptosis and EMT. Next, other levels of regulation, such as the regulation of ferroptosis at the protein level, are also worth investigating. Current studies have focused on transcriptional regulations and epigenetic modifications, with only a few studies describing the molecular mechanisms by which protein autophagy regulates ferroptosis [15], while other molecular biochemical alterations, such as protein post-translational modifications (PTMs), have been less well studied in ferroptosis.

## Figures and Tables

**Figure 1 ijms-24-14614-f001:**
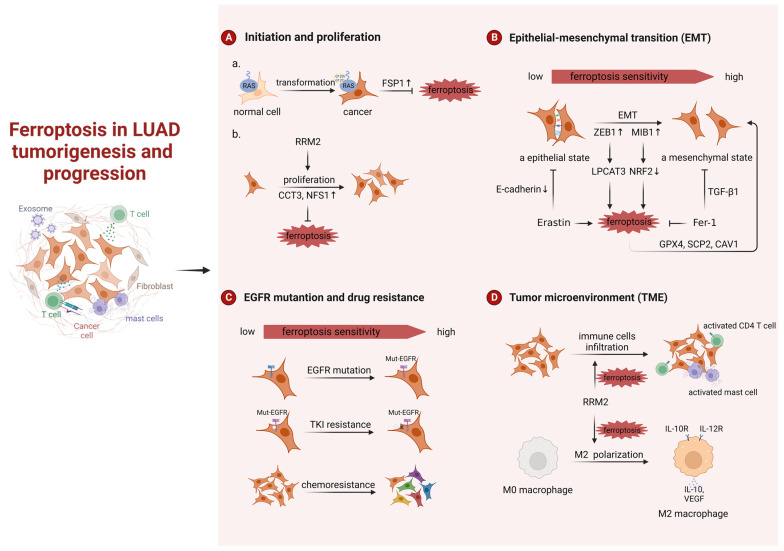
Ferroptosis in LUAD tumorigenesis and progression. a. Ferroptosis in LUAD initiation; b. Ferroptosis in LUAD proliferation. RRM2, ribonucleotide reductase subunit M2; CCT3, chaperonin containing TCP1 subunit 3; NFS1, nitrogen fixation 1; ZEB1, zinc finger E-box binding homeobox 1; Fer-1, ferrostatin-1; TKI, tyrosine kinase inhibitor. Created with BioRender.com.

**Figure 2 ijms-24-14614-f002:**
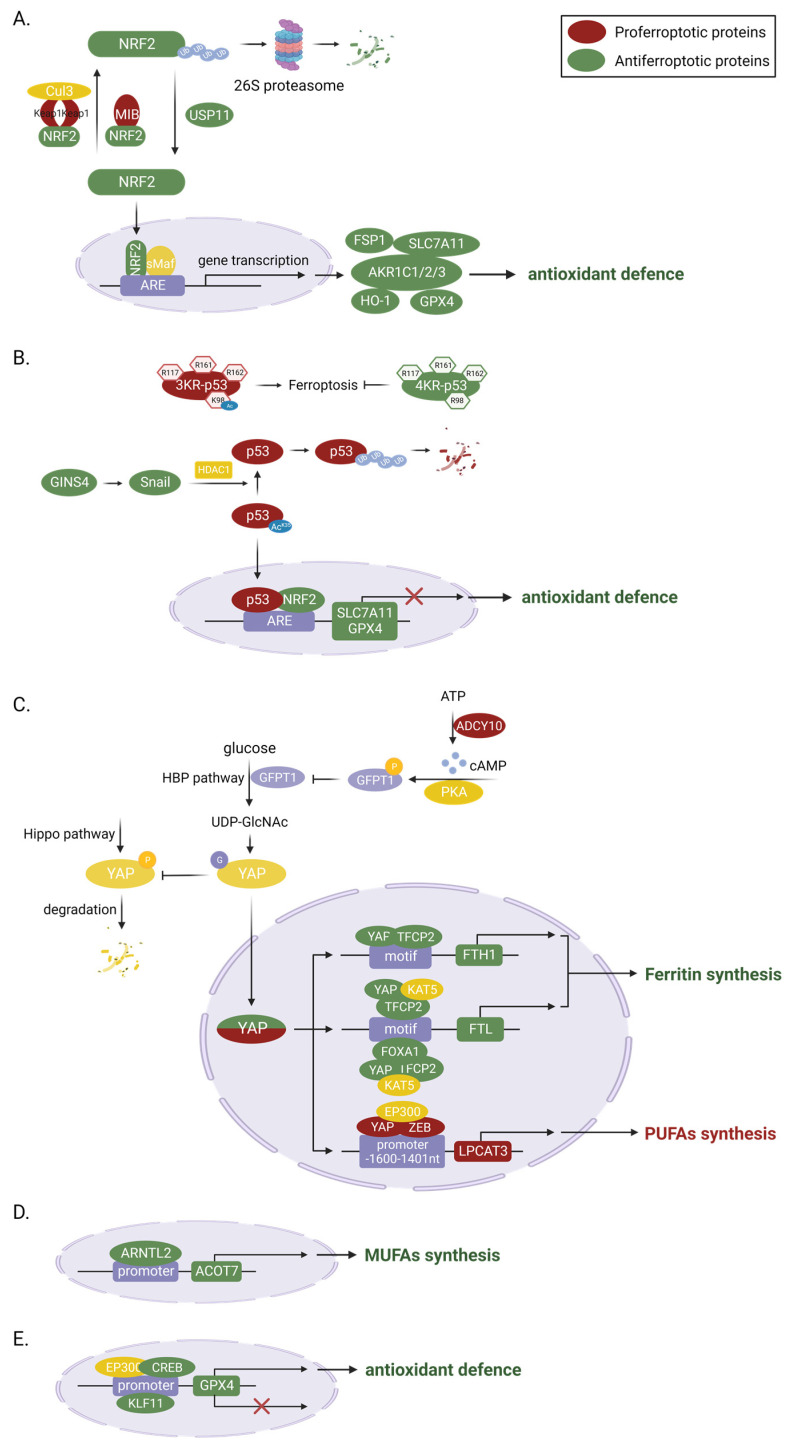
Master transcriptional regulators of ferroptosis in LUAD. (**A**) NFE2L2 is a crucial anti-ferroptotic transcription factor in LUAD cells, as it regulates multiple genes involved in antioxidant defense. The expression of NFE2L2 is primarily controlled by ubiquitination and proteasome-mediated degradation. Multiple E3 ubiquitin ligases or de-ubiquitinating enzymes regulate the sensitivity to ferroptosis by controlling the expression of NFE2L2 in LUAD cells. (**B**) TP53 inhibits ferroptosis and LUAD progression by suppressing the expression of SLC7A11 and GPX4. Acetylation is very important in the regulation of its function in ferroptosis because it increases the stability of the p53 protein. (**C**) YAP1 plays a dual role in ferroptosis, depending on the expression of its target genes. The O-GlcNAcylation of YAP1, which is mediated by O-GlcNAc, enhances the activity of YAP1 and its transcriptional ability. In LUAD cells, the sensitivity to ferroptosis can be regulated by controlling its upstream ADCY/PKA/HBP axis. (**D**) ARNTL2 promotes ferroptosis by promoting the expression of ACOT7, a MUFA synthesis-related enzyme. (**E**) CREB1 inhibits while KLF11 promotes ferroptosis by binding to the promoter of GPX4. Created with BioRender.com.

## Data Availability

Not applicable.

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
