# Peer review of "Regulation of Ferroptosis in Lung Adenocarcinoma"

_ijms, 2023, doi:10.3390/ijms241914614_

Round 1
Reviewer 1 Report
This review about ferroptosis on molecular biology dependent steps and regulation is well done.
But change in proteins of p53, Histone aggregation are mostly chemical radical induced pathways depending on the radical metabolism of cancer cells.
Additional comments
The novelty is a detailed review of ferroptosis. Reading it from the start it is easily to souk in this specific theme because modern strategies, especially in cancer.
I´m coming from the oxidative stress side where chemical reactions, like lipid peroxidation takes place with ferrum or a dirct interaction with proteins, which can be seen at the modification of histones, and down - ad up regulations. So, for me it was a interesting journey and may I will use some techniques to involve this in my research.
Author Response
Thank you for your constructive and kind suggestions.
Our responses to your comments:
- This review about ferroptosis on molecular biology dependent steps and regulation is well done.
Response: Thank you for your positive opinion.
- But change in proteins of p53, Histone aggregation are mostly chemical radical induced pathways depending on the radical metabolism of cancer cells.
Response: Thank you. We did the revision as you suggested.
Additional comments
- The novelty is a detailed review of ferroptosis. Reading it from the start it is easily to souk in this specific theme because modern strategies, especially in cancer.
I´m coming from the oxidative stress side where chemical reactions, like lipid peroxidation takes place with ferrum or a dirct interaction with proteins, which can be seen at the modification of histones, and down - ad up regulations. So, for me it was a interesting journey and may I will use some techniques to involve this in my research.
Response: Thank you.
Reviewer 2 Report
The manuscript by Wei et al. shed light on a very important aspect of Lung Adenocarcinoma therapy. The authors have correctly identified that Ferroptosis, an iron-dependent regulated cell death, caused by lipid peroxidation, has attracted much attention recently as an alternative target for apoptosis in LUAD therapy. They have provided enough background of LUAD and its general mechanism. They have also discussed Lipid peroxidation in details. Then they discussed in details the overview on ferroptosis mechanisms, how its is regulated in LUAD, and application of targeting ferroptosis for LUAD therapy.
Overall, this review is of comprehensive nature, very timely and provides very vital information about the state of the art in the LUAD therapy. This reviewer compliments the authors for such an extensive review and recommends its publication in IJMS.
Minor editing is needed to improve the quality of English in the manuscript.
Author Response
Thank you for your nice advice.
Our responses to your comments:
The manuscript by Wei et al. shed light on a very important aspect of Lung Adenocarcinoma therapy. The authors have correctly identified that Ferroptosis, an iron-dependent regulated cell death, caused by lipid peroxidation, has attracted much attention recently as an alternative target for apoptosis in LUAD therapy. They have provided enough background of LUAD and its general mechanism. They have also discussed Lipid peroxidation in details. Then they discussed in details the overview on ferroptosis mechanisms, how its is regulated in LUAD, and application of targeting ferroptosis for LUAD therapy.
Overall, this review is of comprehensive nature, very timely and provides very vital information about the state of the art in the LUAD therapy. This reviewer compliments the authors for such an extensive review and recommends its publication in IJMS.
Response: Thank you for your very nice advice.
Reviewer 3 Report
The authors reviewed the regulation of ferroptosis in lung cancer. For this purpose, they have compiled data from the literature; their own data are not included. The introduction introduces the topic. Adenocarcinomas of the lung have a very poor prognosis. Modulation of ferroptosis could be a therapeutic option. Ferroptosis is presented in the following: Iron metabolism, lipid metabolism and the antioxidant system are systematically presented based on the literature. This is followed by discussions of the potential role of ferroptosis in lung cancer, both in tumorigenesis and tumor progression. The relationships are presented in a very informative graphic format, but the type is often very small. The multiple regulatory mechanisms open up many potential therapeutic options. Again, the font in the figure is much too small. The authors conclude that new therapeutic principles are possible here. An extensive bibliography concludes the paper.
Author Response
Thank you for your constructive and kind suggestions.
Our responses to your comments:
The authors reviewed the regulation of ferroptosis in lung cancer. For this purpose, they have compiled data from the literature; their own data are not included. The introduction introduces the topic. Adenocarcinomas of the lung have a very poor prognosis. Modulation of ferroptosis could be a therapeutic option. Ferroptosis is presented in the following: Iron metabolism, lipid metabolism and the antioxidant system are systematically presented based on the literature. This is followed by discussions of the potential role of ferroptosis in lung cancer, both in tumorigenesis and tumor progression. The relationships are presented in a very informative graphic format, but the type is often very small. The multiple regulatory mechanisms open up many potential therapeutic options. Again, the font in the figure is much too small. The authors conclude that new therapeutic principles are possible here. An extensive bibliography concludes the paper.
Response: For you reminding, we cited our own study in reference 69. In addition, we have enlarged the font both in figures and table 1, to make them more clear and readable. Thank you for your nice suggestions.